# Viromics unveils extraordinary genetic diversity of the family *Closteroviridae* in wild citrus

**Qiyan Liu[1], Song Zhang[1], Shiqiang Mei[1], Yan Zhou[1], Jianhua Wang[2], Guan-Zhu Han[2], Lei Chen[3], Changyong Zhou[1]\*, Mengji Cao[1]\***

**1** National Citrus Engineering and Technology Research Center, Citrus Research Institute, Southwest University, Beibei, Chongqing, China, **2** Jiangsu Key Laboratory for Microbes and Functional Genomics, Jiangsu Engineering and Technology Research Center for Microbiology, College of Life Sciences, Nanjing Normal University, Nanjing, Jiangsu, China, **3** Industrial Crop Workstation of Xinping County, Yuxi, Yunnan, China

☯ These authors contributed equally to this work.
* zhouchangyong@cric.cn (CZ); caomengji@cric.cn (MC)

## Abstract

Our knowledge of citrus viruses is largely skewed toward virus pathology in cultivated orchards. Little is known about the virus diversity in wild citrus species. Here, we used a metatranscriptomics approach to characterize the virus diversity in a wild citrus habitat within the proposed center of the origin of citrus plants. We discovered a total of 44 virus isolates that could be classified into species *Citrus tristeza virus* and putative species citrus associated ampelovirus 1, citrus associated ampelovirus 2, and citrus virus B within the family *Closteroviridae*, providing important information to explore the factors facilitating outbreaks of citrus viruses and the evolutionary history of the family *Closteroviridae*. We found that frequent horizontal gene transfer, gene duplication, and alteration of expression strategy have shaped the genome complexity and diversification of the family *Closteroviridae*. Recombination frequently occurred among distinct *Closteroviridae* members, thereby facilitating the evolution of *Closteroviridae*. Given the potential emergence of similar wild-citrus-originated novel viruses as pathogens, the need for surveillance of their pathogenic and epidemiological characteristics is of utmost priority for global citrus production.

## Author summary

Closterovirids are principal plant pathogens for citrus trees and other plants, as they sometimes cause new or re-emerging diseases. However, the closterovirid diversity in natural plant hosts, especially citrus plants, is unclear. Here, we describe three novel species and *Citrus tristeza virus* within the family *Closteroviridae* that were sampled from wild citrus trees growing in their natural habitat in southwestern China. The presence of three different taxon classes of the family *Closteroviridae* indicates the geographical uniqueness of the sampling region for citrus closterovirid evolution. Our analysis shows that frequent horizontal gene transfer, gene duplication, alteration of expression strategy, and

**Data Availability Statement:** All relevant data are within the manuscript and its Supporting Information files. The virus genomic sequences

also could be accessed in ncbi.nlm.nih.gov/ with accession number MW365399–MW365403.

**Funding:** This research was supported by the National Key R&D Program of China to MC (2019YFD1001800), National Natural Science Foundation of China to MC (32072389), Fundamental Research Funds for the Central Universities to MC (XDJK2018AA002), Innovation Program for Chongqing's Overseas Returnees to MC (cx2019013), 111 Project to CZ (B18044), and Earmarked Fund for China Agriculture Research System to CZ (CARS-26-05B), National Training Program of Innovation and Entrepreneurship for Undergraduates to SM (202010635099). The funders had no role in the study design, data collection and analysis, decision to publish, or preparation of the manuscript.

**Competing interests:** The authors have declared that no competing interests exist.

recombination have been important evolutionary processes in the diversification of the family *Closteroviridae*. Our study also shows the significance of natural reserves as potential sources of disease agents endangering cultivated crop plants.

## Introduction

Characterization of plant viruses has generally focused on those causing symptoms in cultivated plants. Through the use of viral metagenomics (viromics) techniques, many new viruses have been discovered, greatly enlarging our understanding of the ecology and evolution of plant viruses in nature [1–3].

Members of the family *Closteroviridae* have long, helical, filamentous particles with positive-sense single-stranded RNA genomes. They infect a wide range of agriculturally important crops, causing severe economic damage [4]. The family includes four genera: *Closterovirus* (with aphid-transmitted members), *Ampelovirus* (with mealybug/soft-scale-transmitted members), *Crinivirus* (with whitefly-transmitted members), and *Velarivirus* (without known vectors). Viruses of the genus *Crinivirus* have bi- or tripartite genomes that are separately encapsidated, whereas viruses belonging to *Closterovirus*, *Ampelovirus*, and *Velarivirus* have monopartite genomes [4].

The viral genomes of closterovirids contain two characteristic genomic blocks. The first is the replication-related gene block that includes the methyltransferase (Mtr) and helicase (Hel) domains encoded by open reading frame (ORF) 1a and the RNA-dependent RNA polymerase (RdRP) domain encoded by ORF1b. The second gene block encodes five proteins that function in virion assembly and transport, namely a small hydrophobic protein, a heat shock protein homolog (HSP70h), a ~60-kDa protein, the major capsid protein (CP), and a duplicated version of the latter (the minor capsid protein, CPm). Apart from the two conserved gene blocks, additional nonconserved genes encode proteins that vary in number, arrangement, function, and origin within and between genera, with most having no detectable similarity with other viruses [4,5]. Some of those proteins have important functions, including suppression of plant RNA silencing, RNA repair, and host range broadening [6], whereas the functions of others are still unknown.

Homologous recombination, capture of foreign genes (e.g., those encoding protease and HSP70h domains), and duplication of intragenomic sequences (e.g., CPm) have shaped the extraordinary molecular and biological diversity of the family *Closteroviridae* [7]. A scheme has been proposed for the genome evolution of the family *Closteroviridae*, whereby a common ancestor with the RdRP, a p6-like movement protein and a single CP acquired other modules during the course of evolution [8].

The aphid-borne citrus tristeza virus (CTV), the only known citrus-infecting closterovirid, causes the quick decline, seedling yellow, stem pitting, and slow deterioration that have been associated with long-term chronic losses in commercial citrus production [9]. This highly destructive viral pathogen has resulted in numerous disease outbreaks in most citrus-growing regions of the world [10].

The genus *Citrus* L. is considered to have originated from the southeast piedmonts of the Himalayas, a region that includes Yunnan Province of China [11]. As a part of the Himalayan biodiversity hotspot, the mountainous Yunnan is one of the botanically most diverse regions in the world [12]. Its peaks and deep valleys are potential barriers to species spread, and its climatic, geologic and topographic diversity provides an ideal environment for the formation and development of the flora [13]. The Ailao Mountains, located in central and southern

Yunnan Province and ranging in elevation from 100 to 3,000 m above sea level, preserve the largest, most continuous natural subtropical evergreen-broadleaf forest in China [14].

Due to the limits of technology, early research in wild citrus plant pathology only analyzed molecular characteristics of incomplete genomic sequences of CTV [15,16]. Since metagenomics has been applied to the discovery and identification of novel viruses, our knowledge of the genome and epidemiology of citrus-infecting viruses has expanded tremendously [17–19]. However, the research has mainly focused on viral mechanisms and host pathology. While more than 30 citrus virus and viroid diseases have been identified and characterized [20], little is known about virus diversity and ecology in wild ecosystems. In particular, because we know that grapevine, cherry, and blackberry can be infected by more than one closterovirids [4], there is a need to better understand the diversity of their citrus counterparts. Here we investigated the diversity of closterovirids in wild citrus trees in the Ailao Mountain region using ribosomal RNA (rRNA) depleted transcriptomics. Three novel citrus-infecting closterovirids and different CTV isolates were identified and characterized, providing new insights into the virus ecology in wild citrus plants and the evolution and genetic diversity of the family *Closteroviridae*.

## Results

### Discovery of closterovirids in wild citrus

A total of 49 samples were collected in a wild citrus habitat (Fig 1), and 16 libraries were prepared (S1 Table) and subsequently sequenced. *De novo* assembled contigs were classified as belonging to CTV and three novel closterovirids (Fig 2, S1 Data). One of the viral contigs was identified as originating from a novel closterovirid that shared similarities with different members of the family, but it did not correspond to any established genus (S1A and S1B Fig). The novel virus was tentatively named citrus virus B (CiVB). The two other novel viruses, shared sequence similarity with members of the genus *Ampelovirus* (S1C Fig) and were tentatively

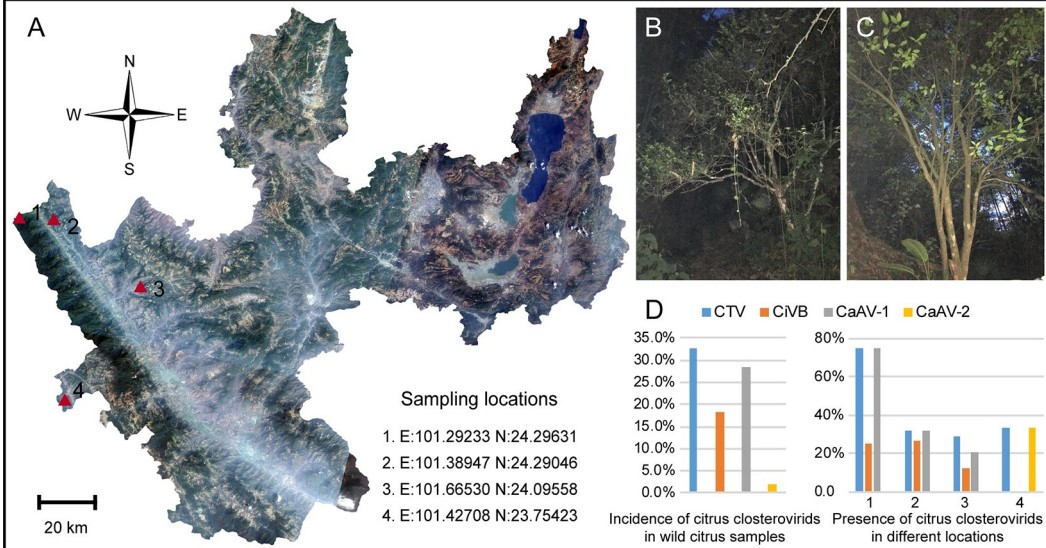

**Fig 1. Geography, habitat, and virus incidence of the wild citrus trees we examined.** (A) Sampling locations within the Ailao Mountain area of Yuxi City. Location1, E:101.39668 N:24.29666; location2, E:101.38947 N:24.29046; location3, E:101.66530 N:24.09558; location 4, E:101.42708 N:23.75423. (B, C) Habitat of wild citrus trees. (D) Incidence of citrus closterovirds in wild citrus samples (left) and presence of citrus closterovirids in different sampling locations (right).

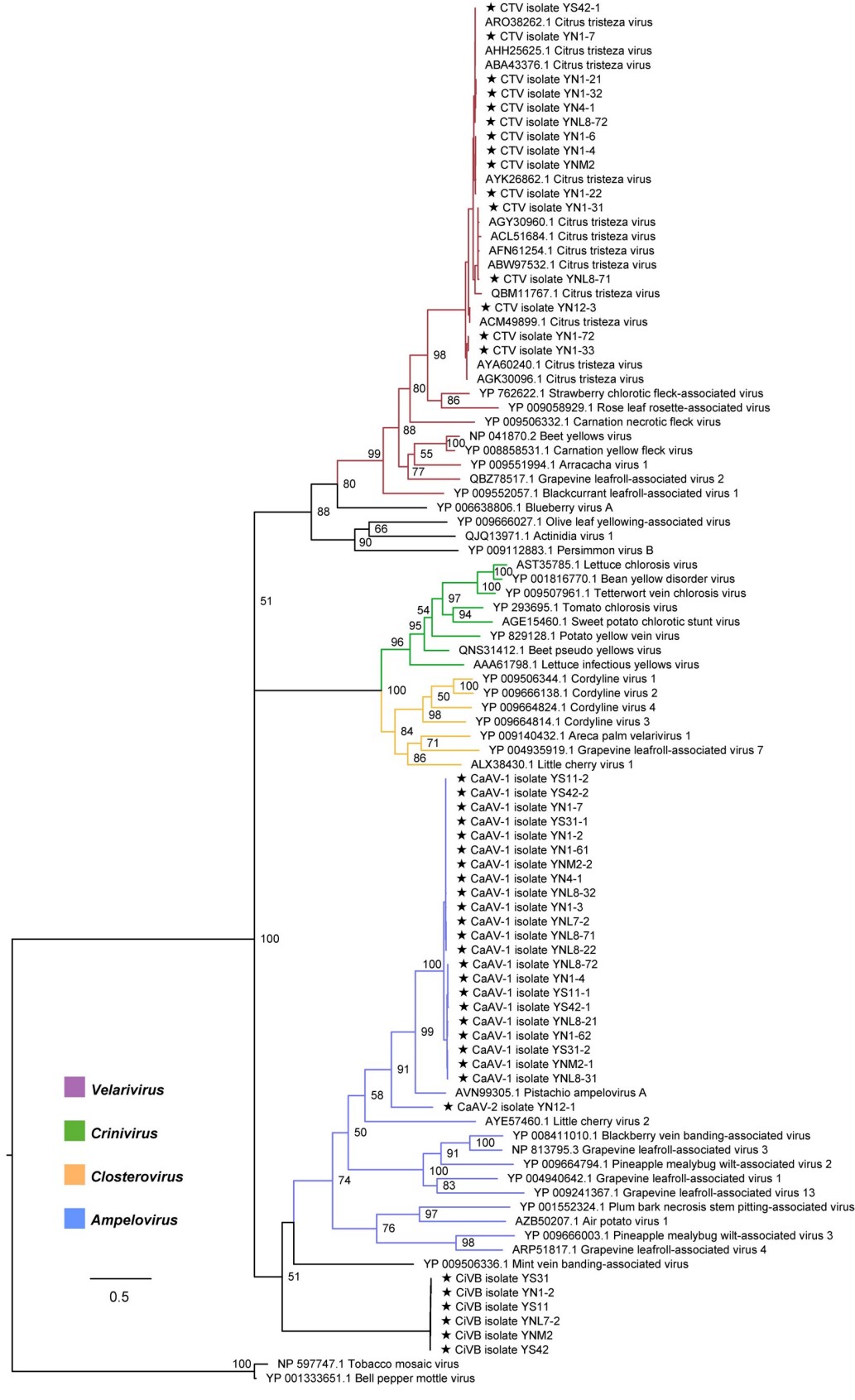

**Fig 2. Phylogenetic tree of the family *Closteroviridae* based on RdRP inferred using ML methods.** ML tree derived from the RdRP gene of newly identified and representative closterovirids. Two tepoviruses were used as an outgroup. Branch support was inferred by bootstrapping with 1,000 replicates. The scale bar represents the number of substitutions per site.

named citrus-associated ampelovirus 1 (CaAV-1) and citrus-associated ampelovirus 2 (CaAV-2). The number of viral contigs reads and average coverage for each sequencing library can be found in S2 Table. In all the sequenced samples, the average coverage for CaAV-1 was much higher than for all other citrus closterovirids. The single contigs from *de novo* assembly corresponded to nearly the complete viral genomes, and RT-PCR and Sanger sequencing using overlapping primers covering the entire genomes did not reveal any breakpoints. Based on these results, we inferred that CiVB, CaAV-1, and CaAV-2 are monopartite rather than multipartite viruses.

In a total of 49 wild citrus samples, the incidence of CaAV-1 (14/49, 28.6%) was similar to CTV (16/49, 32.7%), both higher than CiVB (9/49, 18.4%). CaAV-2 was only detected in one sample in location 4 (Fig 1D). Mixed infection was detected in 16 of 49 samples (32.7%), in which all CiVB isolates were mixed with CaAV-1, CTV or both (S3 Table). After eight months of grafting, all surviving grafted plants were positive for the corresponding infected novel closterovirids, indicating that CiVB, CaAV-1, and CaAV-2 had the capacity to be graft-transmitted to citrus trees.

Among the collected wild citrus samples, the sample YN8-2, which was only infected by CaAV-1 without any other viruses or viroids, exhibited leafroll symptoms, as shown in S2A Fig. As all CiVB isolates showed mixed infection with other citrus closterovirids, it was difficult to analyze their biological symptoms. After eleven months of grafting, three Morocco sour orange [*Citrus*. *aurantium*] plants inoculated with YN8-2 and that were CaAV-1 positive exhibited a series of symptoms (S2B Fig), with the leaf margin becoming irregular and the leaf blade upward or down curling. The barks of CaAV-1 positive sour orange was subsequently grafted on a Duncan grapefruit (*C. paradisi*) which exhibits boat-shaped leaf curling (S2C Fig) after five months of grafting.

## Genomic organization of newly-identified closterovirids

Three novel citrus closterovirids, CiVB isolate ZLV7-2, CaAV-1 isolate YN1-62, and CaAV-2 isolate YN12-1 were examined for genomic features. Isolate YN1-6 of CTV was used for comparison purposes with the new citrus closterovirids. Only the genomic features identified in all isolates were considered reliable. Furthermore, the mapping reads of CTV isolates were preferentially distributed at the 3'-terminal region of the genomic RNA (Fig 3A), similar to the observation of those viral small RNAs of 21–24 nt of CTV [21].

From six samples, a novel monopartite virus, CiVB (GenBank accession no. MW365399) was identified, with some of its potentially encoded proteins sharing the highest amino acid sequence identity with those of different closterovirids (S1B Fig). Six CiVB isolates shared 98.2% to 99.7% genomic nucleotide identity. The CiVB genome comprises 16,957 nucleotides (nt) with 13 ORFs (Fig 3B). ORF1a putatively encodes a polyprotein with leader proteinase (L-Pro), Mtr (l03298), and Hel domains (pfam01443). Two transmembrane alpha-helices (TMHs) connected by hydrophilic loops were identified in the variable central region of ORF1a. ORF1b putatively encodes RdRP (with conserved domain RdRP-2, pfam00978). ORF2 encodes p8a with a predicted signal peptide that has a TMH. ORF3, ORF5, ORF6 and ORF7 encode HSP70h, 58-kDa protein (p58), CP, and CPm, respectively. Small ORF4 and five ORFs downstream of the CPm (ORFs 8–12) encode the putative proteins p8b, p9a, p9b, p34a, p11,

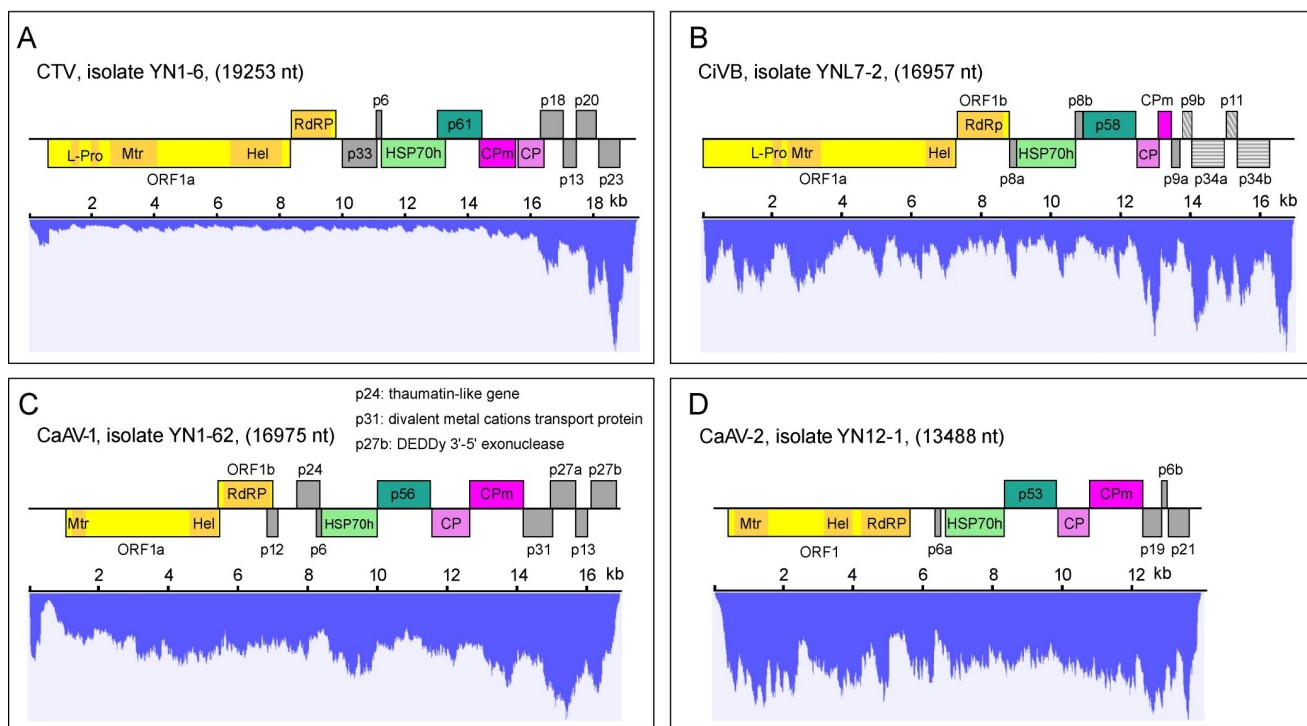

**Fig 3. Genomic organization and transcriptome mapping analysis for citrus closterovirids.** Genomic organization and transcriptome mapping of CTV (A), CiVB (B), CaAV-1 (C), and CaAV-2 (D). The deeper yellow shade in the C-terminal of genomes represents the location of the conserved domains. The p9b and p11, p34a and p34b of CiVB that labeled in the same stripe shapes in panel B represent the identified gene duplication. The p24, p31, and p27b of CaAV-1 labeled in panel C represent identified horizontally transferred genes. L-Pro, leader proteases; Mtr, methyltransferase; Hel, helicase; RdRP, RNA-directed RNA polymerase; HSP70h, heat shock protein 70 homolog; CP, major coat protein; CPm, minor coat protein.

and p34b. BLAST and CD-based searches did not reveal any statistically significant hits for these proteins in the databases.

From 12 samples, we identified 18 isolates of a novel monopartite ampelovirus, CaAV-1 (GenBank accession no. MW365401), with 86.7–99.3% genomic nucleotide identity. The complete genomic sequence of the CaAV-1-YN1-62 consists of 16,975 nt with 13 putative ORFs (Fig 3C). ORF1a encodes a putative polyprotein containing the signatures of Mtr (cl03298) and Hel (cl26263). ORF1b putatively encodes RdRP (pfam00978). ORF2 encodes p12 with two TMHs. ORF3 encodes p24 with a thaumatin-like proteins domain (TLP, cd09218, E-value = 3.85e-06). ORF4 encodes p6 that contains a TMH. ORF5, ORF6, ORF7, and ORF8 encode HSP70h, 56-kDa protein (p56), CP, and CPm, respectively. ORF9 and ORF12 encode p31 and p27b, which have specific hits for divalent metal cations (Fe/Co/Zn/Cd) transport protein (FieF superfamily, cl30791, E-value = 5.11e-06) and DEDDy 3'-5' exonuclease of WRN class (WRN_exo, cd06141, E-value = 5.59e-40) domains, respectively. ORF10 and ORF11 encode p27a and p13, neither of which shared statistically significant identity in the database.

From sample YN12-1, one large contig of 13,488 nt was recognized as originating from another novel ampelovirus, CaAV-2 (GenBank accession no. MW365402). The complete sequence of the CaAV-2 genome consisted of 13,407 nt and potentially encompassed 10 ORFs (Fig 3D). ORF1 encodes a polyprotein with Mtr (cl03298), Hel (cl26263) and RdRP (pfam00978) domains. Intriguingly, CaAV-2 encodes RdRP within ORF1 together with other replication-associated genes; this pattern differs from all known closterovirids, which encode RdRP gene in an ORF1b via a +1 ribosomal frameshift strategy [4]. ORF2 encodes p6a that has

a TMH domain. ORF3, ORF4, ORF5 and ORF6 encode a series of conserved proteins, namely HSP70h, p53, CP, and CPm, respectively. The remaining three ORFs downstream of the CPm, namely, ORFs 7, 8, and 9, encode the putative proteins p19, p6b, and p21, respectively, with no shared identity in the databases. The p6b has a TMH domain.

## Phylogenetic analysis of CiVB, CaAV-1 and CaAV-2

The maximum likelihood (ML) trees based on the RdRP, CP, and HSP70h segment sequences produced similar clustering groups with the classified genera [4]. *Crinivirus* and *Velarivirus* clustered together with *Closterovirus* in the RdRP and HSP70h trees, while they formed monophyletic groups in the CP tree. The phylogenetic discordance was observed in the tanglegram of RdRP and CP deriving from the representative members of the family *Closteroviridae*, which showed that recombination events have not been rare during the evolutionary course of this viral family (Figs 4 and S1A).

CiVB formed a separate group in both the tanglegram and the phylogenetic tree of the HSP70h gene (Figs 4 and S1A). With low and similar amino acid sequence identity of the polyprotein, RdRP, HSP70h, and CP with extant genera of the family *Closteroviridae* (S1B Fig), CiVB may represent a new genus of the family *Closteroviridae*. CaAV-1 and CaAV-2 clustered with the genus *Ampelovirus* in all phylogenetic trees. Indeed, CaAV-1 formed a subcluster with Pistachio ampelovirus A (PAVA) for all genes (Figs 4 and S1A). CaAV-2 showed interspecies recombination evidence since it clustering with CaAV-1 and PAVA in RdRP and CP gene trees while clustering with grapevine leafroll-associated virus 13 in Hsp70h gene trees. The amino acid sequence identities of the RdRP, HSP70h, and CP proteins shared respectively between CaAV-1, CaAV-2 and the ampeloviruses differed by more than 25% (S1C Fig). Thus, these two viruses might represent novel species in the genus *Ampelovirus*.

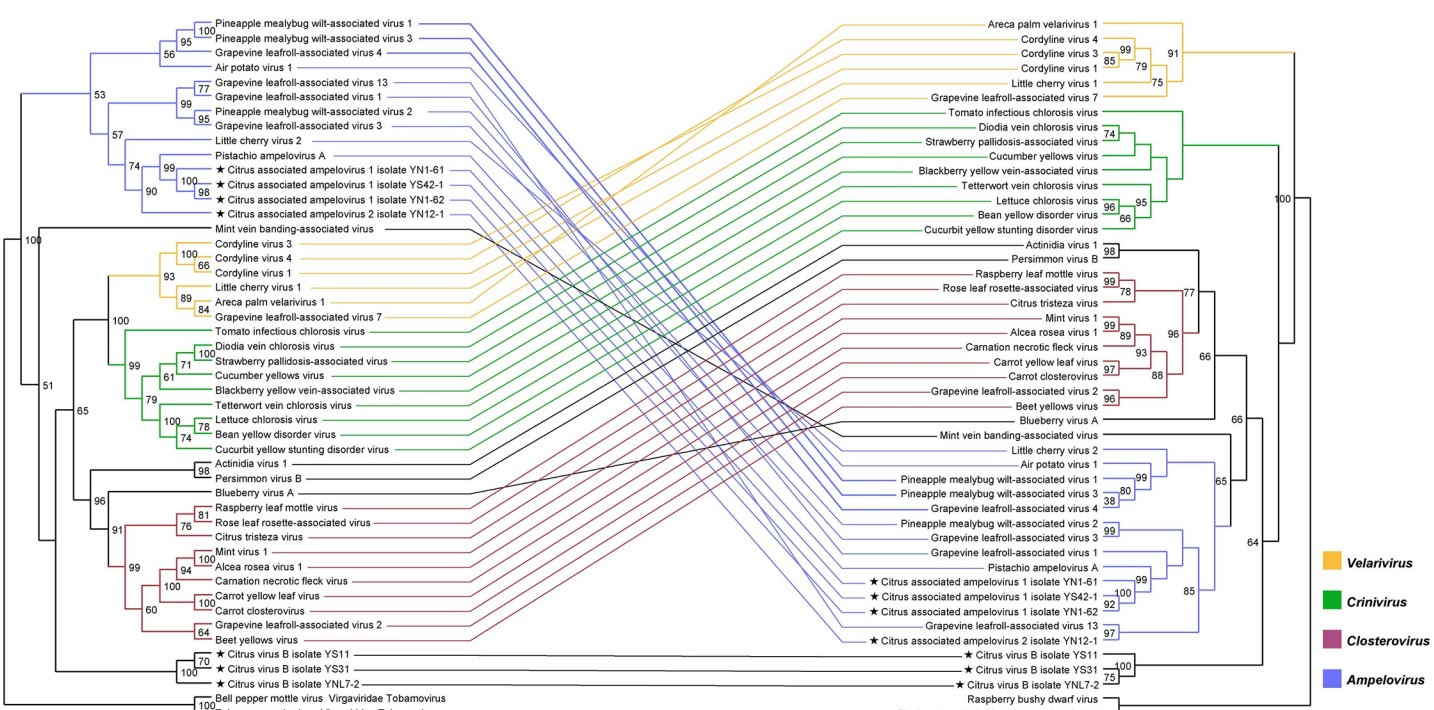

**Fig 4. Co-evolutionary analysis for the family *Closteroviridae*.** Tanglegram of ML analyses of newly-identified and known closterovirids based on the amino acid sequence of RdRP and CP. Branch supports were inferred by bootstrapping with 1000 replicates. Two tobamoviruses and two idaeoviruses were respectively used as outgroups of RdRP and CP phylogenetic trees.

## Formation of genomic complexity of the family *Closteroviridae*

Three genes with sequence similarity to other organism genes were identified in all CaAV-1 genome isolates (Fig 3C). The genes encoding TLP were systematically identified at different genomic regions of seven closterovirids that belonged to different taxa (Fig 5A), thus highlighting the transferability of this kind of protein in *Closteroviridae* evolution. The host defense and developmental processes related TLP gene exists widely in plants, fungi, nematodes and insects, and the closteroviral homologs showed affinity to fungal TLPs (Fig 5B). Genes coding for a divalent metal cation transporter and a DEDDy 3'-5' exonuclease (p31 and p27b in CaAV-1, p31 and p25 in PAVA) appeared in the same locations of the CaAV-1 and PAVA genomes (GenBank accession no. MF198462) (Fig 5A). The DEDDy 3'-5' exonuclease was also identified in coronaviruses with functions of reducing replication errors, proofreading, and

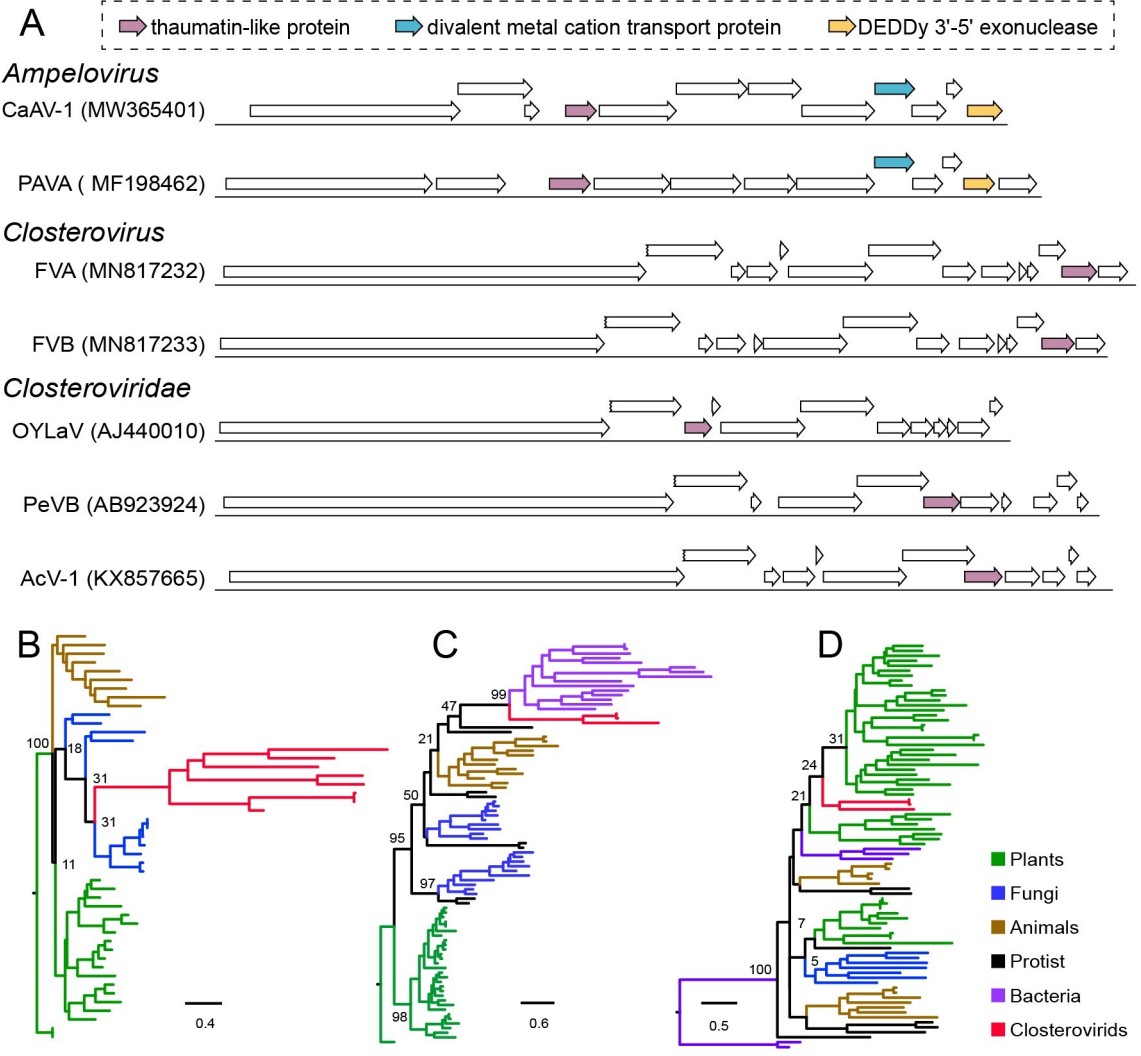

**Fig 5. HGT analysis of three genes of closterovirids.** (A) Locations of genes encoding TLP, divalent metal cations transporter, and DEDDy 3'-5' exonuclease in CaAV-1 and other closterovirids genomes. The unrooted ML trees derived from TLP (B), divalent metal cations transporter (C) and DEDDy 3'-5' exonuclease (D) genes of closterovirids and their representative homologs. The supporting values on major branch was inferred by bootstrapping with 1,000 replicates. The scale bar represents the number of substitutions per site. CaAV-1, citrus associated ampelovirus 1; PAVA, pistachio ampelovirus A; FVA, fig virus A; FVB, fig virus B; OYLaV, olive leaf yellowing-associated virus; PeVB, persimmon virus B; AcV-1, actinidia virus 1.

native recombination mediated activities [22,23]. The gene coding for metal cation transporter and a DEDDy 3'-5' exonuclease clustered with their bacteria and plant homologs, respectively (Fig 5C and 5D). The bootstrap values of phylogenetic trees for these genes were relatively low. However, under the rapid evolution background of RNA viruses, these genes still showed certain sequence similarities (S4 Fig). Thus, these genes appeared to be derived from horizontal gene transfer (HGT).

Genes that were located in the C-termini of CiVB and encoded putative p9b, p34a, p11, and p34b had no statistically significant hits in the public databases. Employing a local BLAST approach, we detected two cases of gene duplication in these four genes that formed a tandem repeat (S3A Fig). Specifically, there was 25.49% amino acid sequence identity between p9b and p11 (E-value = 8.88E-07), and 48.71% amino acid sequence identity between p34a and p34b (E-value = 9.20E-110).

The phylogenetic reconstruction of the CP, CPm and p60~like genes of the family *Closteroviridae* supported p60 presence in the common ancestor of the family, and the CPm gene of the genus *Ampelovirus* and *Closterovirus* suggests duplication of their CP genes independently (S3B Fig), similar to the results of a previous study [5]. The CPm genes of CiVB, criniviruses and velariviruses clustered together and were separate from their respective CP genes, indicating that their CPm genes were not derived from independent gene duplication events.

## Genotyping, recombination, and asymmetrical genomic variation of CTV and CaAV-1 in wild citrus

The Neighbor-net analysis showed that 22 CaAV-1 isolates were segregated into two different genotype classes (Fig 6A). Fifteen CTV isolates were assigned to CTV genotypes VT, T36, T3,

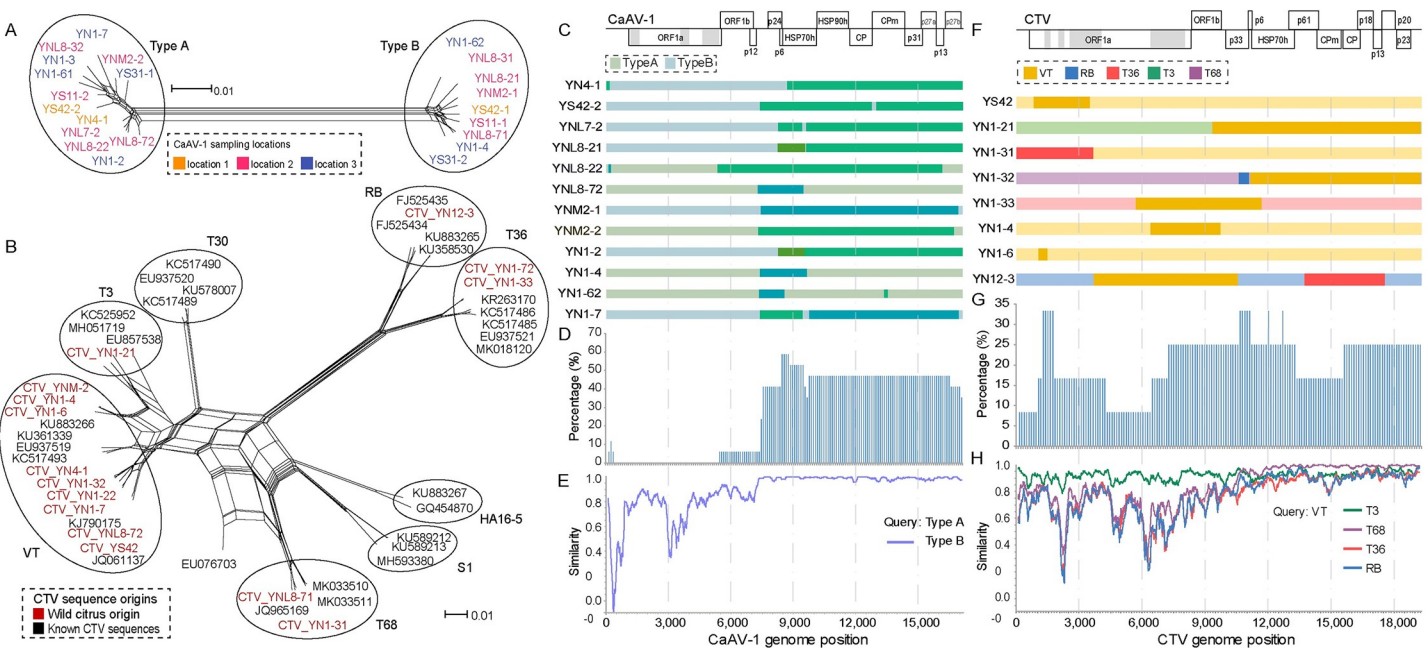

**Fig 6. Genotyping, recombination, and asymmetrical genomic variation of CTV and CaAV-1 in wild citrus.** Neighbor-net generated from the complete genome sequences of different CTV isolates (A) and CaAV-1 isolates (B). Parallel paths among the different viral isolates represent putative recombination events. (C, F) Maps of recombination patterns and parental lineages. The different color schemes depict different CaAV-1 and CTV lineages. The major parents are in a light shade, and the minor parents are in a deeper shade. (D, G) Rate of recombinant isolates among total isolate numbers in a sliding 100 nucleotide window. The *y*-axis represents the proportion of recombinant sites. (E, H) The sequence identity of CaAV-1 and CTV. The map was made using SimPlot. The *y*-axis represents the sequence identity. The *x*-axis represents CaAV-1 and CTV genome positions.

T68, and RB genotype classes that covered almost all ancestors of the CTV genotypes detected in commercial citrus crops. The CTV isolates from wild citrus clustered together with their counterparts from cultivated citrus without demarcation in the neighbor-net analysis (Fig 6B). Thus, the different CTV genotypes appear to have multiple origins. From the viromics analysis, mixed infection of different genotypes existing within a single sample as higher in CaAV-1 (8/14, 57.1%) than in CTV (4/10, 40%).

The reticulation of the neighbor-net analysis indicated potential recombination events both in CTV and CaAV-1, and the discordant phylogenetic topology further supported the recombination signals. As shown in S5 and S6 Figs, the replication-related module tree was in accord with the neighbor-net analysis, in which the same genotypes clustered together, while the HSP70h and CP trees are largely different. Recombination in CaAV-1 seemed to display a bias, occurring more frequently in the regions downstream of the replication-associated module. The recombinant minor parents (marked with deep colors in Fig 6C) mainly come from genotype A. Of the 12 recombinants detected in the 22 CaAV-1 isolates, 8 had large recombined fragments ($>$ 5,000 nt) in the 3'-half portion of the genome (Fig 6C).

The CaAV-1 genotypes shared high sequence identity ($>$94%) throughout the 3'-half of the genome (downstream of the p24), diverging more in the 5'-half where the sequence identity was 76%, a phenomenon we described as asymmetrical genomic variation. This phenomenon has also been reported previously for CTV [24]. The CTV isolates identified here shared more than 88% sequence identity in the 3'-half portion of the genome (downstream of p33) but only 72% sequence identity in the 5'-half counterpart (Fig 6). Specifically, we identified two special examples in samples YNL8-2 and YNL 8–7, each coinfected with two CaAV-1 genotypes. In these two samples, two CaAV-1 isolates had diverse 5'-half genome sequences but shared similar 3'-half genome sequences. As shown in the left penal of S7 Fig, different CaAV-1 isolates from the same samples (YNL 8–2 and YNL 8–7) shared the same reads (in yellow) at the 3'-half genome portion. Single nucleotide polymorphisms (SNPs) did not contribute to that genome asymmetrical variation, as the Spearman correlation analysis of the association between the lowest nucleotide sequence identity and the number of SNPs detected no correlation for either CTV (r = 0.198) or CaAV-1 (r = 0.143) (right penal of S7 Fig).

## Discussion

From one isolated region of wild citrus, we found viruses that belonged to the known genus *Ampelovirus* (CaAV-1 and CaAV-2, two putative new species), *Closterovirus* (CTV), and CiVB, which may represent a new genus in the family *Closteroviridae*. The viruses newly identified in this study show that the diversity of the family *Closteroviridae* exceeds what was previously thought, since the viruses reported here: i) are highly divergent compared to known members of the family; ii) have an extraordinary ability to recombine with different sources and, iii) vary widely in their genome organization and expression strategy.

Why are there so many closterovirids in wild citrus species but not other common viruses? This may be because while CTV and perhaps the ancestor of other closterovirids are native to citrus plants, other viruses may have been transferred to cultivated citrus crops from other plants. This idea should be further explored by examining wild citrus growing in the same areas as cultivated varieties. Otherwise, extant members of the family *Closteroviridae* are known to be transmitted in cultivated areas by a few specific insect vectors, whereas in a natural environment the spectrum of such vectors may be wider. Thus, in natural citrus habitats that have had no human intervention, the insect-borne closterovirids may infect wild citrus plants over a relatively wide area compared with other viruses. In addition, the citrus viruses perhaps had multiple origins, and other citrus viruses and viroids that originated in other

areas have not arrived here as the Ailao Mountains are to some extent isolated from the rest of the citrus-planted regions.

Within the long co-evolutionary period, these closterovirids have co-existed with other parasites such as fungi and bacteria in their plant hosts that enable viruses to recruit genes via HGT from different sources. Three horizontally acquired genes identified in CaAV-1 and other closterovirids showed affinity to their fungal, bacterial, and plant counterparts. While plant TLPs have been classified as members of the pathogenesis-related protein family, as they are induced and accumulated under biotic or abiotic stress [25,26], the functions of their fungal counterparts are still unknown. The DEDDy 3'-5' exonuclease was identified both in plant closteroviruses and animal corona-like viruses, a result that may illustrate the demand for fidelity enhancing in large RNA genomic replication. As metal ions are required for exonuclease activity [22,23], the metal cation transporter genes of CaAV-1 and PAVA may cooperate with the exonuclease via providing the required metal ions. Given the rapid evolution of the RNA viruses, the tandem repeat gene in the CiVB recognized via amino acid sequence alignment means that the duplication event may not have occurred in the deep past [27]. Further studies of these horizontally acquired genes and duplicate genes should elucidate their evolution and functions. Our results emphasized HGT and gene duplication as important processes in the formation of the genomic size and complexity of the members of the family *Closteroviridae*. Subsequent codivergence and cross-species transmission events may produce the genes with no detectable similarity in extant closterovirids that have diverged independently, facilitating infection of different hosts or transmission by different insect vectors.

While clearly within the clade of ampeloviruses based on sequence similarity and phylogenetic analysis, both metatranscriptome sequencing and Sanger sequencing confirmed that the RdRP gene of CaAV-2 is expressed with other replication-associated genes together within the ORF1 which has no internal stop codon. Thus, CaAV-2 seems to employ a unique strategy to express its RdRP gene; this awaits further experimental study. A T7 transcript encoding the CaAV-2 ORF1 product expressed *in vitro* with isotope labeling would be feasible to determine whether there is a potential ribosomal frameshifting [28,29].

All sequenced CTV and CaAV-1 genomes showed asymmetrical variation characteristics that may be derived from recombination events between potential unknown parent viruses that provided the N-terminal fragment and prototypes of CTV and CaAV-1. Otherwise, restricted by the disruption of a series of functions like virion assembly and transmission and host defense response [30,31], the C-terminal fragment has been under strenger purifying selection pressure than the N-terminal fragment.

Viruses are a two-edged sword. As pathogens, they affect diverse agriculture and natural plants and cause host diseases. From a beneficial perspective, a few closterovirus-derived genes and RNA interference vectors have been developed to control plant diseases [32,33]. With their large genomes, the novel citrus viruses may be suitable for construction of new vectors, serving as a tool for citrus tree protection and improvement [9,34].

Some viruses may have coevolved with these wild hosts over a long period, making the latter natural reservoirs for viruses that could cause new or re-emerging diseases. Present-day monocultures over large areas are particularly vulnerable to a wide range of viruses, while globalization has facilitated their transport to new areas. It is in this framework, where wild and cultivated citrus species have come into contact via humans and insect vectors, that CTV became prevalent and destructive for the citrus industry. The novel viruses may have dispersed, as did CTV, from citrus orchards near the Ailao Mountains region. Our findings should serve as a warning that new citrus-infecting closterovirids may exist and may pose new problems for citrus production. Accordingly, we need to continue monitoring and studying these novel viruses and to examine other areas harboring wild citrus plants to provide a

foundation for deploying protective strategies in the future. The lessons from the recent emergence of the covid-19 pandemic apparently following a similar scenario of virus spreading from ecological niches hosting virus species to populations that were never previously exposed need to be paid attention by those concerned with citrus tree health.

## Materials and methods

### Sample collection

This study was based on the analysis of RNA libraries from wild citrus tree samples from the Ailao Mountains region (Fig 1A-C) in central Yunnan Province, southwestern China, collected between July 2018 and August 2019. Some samples from the same location were pooled for library construction and transcriptome sequencing. RT-PCR was used to screen the presence of these novel closteroviruds and CTV using specific primers listed in S4 Table.

Citrus branch samples were grafted onto virus-free seedlings (Morocco sour orange) using four bark patches to preserve the virus materials. All grafted plants were grown in an insect-proof greenhouse. New flush leaves of the grafted seedlings were screened by RT-PCR with viral-specific primers five months after grafting to test whether the viruses were graft-transmissible. The barks of CaAV-1 positive sour orange was grafted on different citrus seedlings of Duncan grapefruit and others to detect the biological features of the new citrus closteroviruds.

### RNA library construction and sequencing

Total RNA was extracted using TRIzol LS (Invitrogen, CA, USA), and rRNA was removed using a Ribo-zero rRNA Removal Kit (Epicentre, WI, USA). The rRNA-depleted RNA libraries were constructed using a NEBNext Ultra Directional RNA Library Prep Kit for Illumina (NEB) and subsequently sequenced on the X-ten platform (Illumina CA, USA), generating paired-end reads of 150 bp. The information concerning library construction and corresponding geographic locations of samples for each sequencing library can be found in S1 Table.

### Sequence assembly and virus discovery

For each library, sequencing read data were processed to remove adapter sequences and low quality reads through the CLC Genomics Workbench 9.5 (Qiagen, CA, USA). The resulting clean data were mapped to citrus genomes [35,36] to filter out the host reads. *De novo* assembly was then performed using the Trinity program (Broad Institute). To identify potential viruses, we compared the assembled contigs against the GenBank database using the BLASTx program.

### Confirmation and extension of virus genomes

Each potential viral genome was further examined using one-step RT-PCR and rapid amplification of cDNA ends (RACE) kit (GeneRacer, Invitrogen, MD, USA) using overlapping primers designed from the assembled sequences (S3 Table). The resulting cDNA products were purified and cloned into the pGEM-T Easy Vector (TransGen Biotech, Beijing, China), which was later transferred into DH5α competent cells (Takara, CA, USA). At least five clones per amplicon were randomly selected and sequenced in both directions by the (BGI, Shenzhen, China).

### Virus genome annotation

For each newly-identified viral genome, ORFs were predicted using the ORF finder (https://www.ncbi.nlm.nih.gov/orffinder/). The potential functions of the encoded proteins were

inferred from the NCBI protein database (https://www.ncbi.nlm.nih.gov/protein) and conserved domain database (https://www.ncbi.nlm.nih.gov/cdd/) using BLAST searches and Batch CD-Search. Relationships of diverse unknown proteins of new and known closterovirids were assessed using local blast within TBtools [37] and were then analyzed using Cytoscape [38,39] with E-value < 1E-5. Transmembrane domains and signal sequences were predicted with SignalP and TMHMM programs on the DTU Health Tech prediction servers (http://www.cbs.dtu.dk/services/). Protein hydropathy was analyzed using ProtParam (https://web.expasy.org/protparam/). The nucleotide and amino acid sequences of viral genomes were calculated with the CLC Genomics Workbench based on MAFFT alignments of the E-INS-I algorithm [40].

## Inference of virus evolutionary relationships

The sequences of RdRP, HSP70h, and CP of newly-identified and known closterovirids and corresponding outgroup sequences were used to construct corresponding ML phylogenetic trees. To confirm the HGT events, the hits sequences in BLASTP program were aligned and analyzed using the FastTree program [41]. Representative protein sequences were selected for constructing ML phylogenetic trees. The sequences were aligned and manually trimmed to remove ambiguously aligned regions of alignments used for evolutionary analysis. The ML phylogenetic trees were inferred using IQ-TREE (version 1.6.12) under the substitution models chosen according to the Bayesian Information Criterion [42]. Topological support was assessed with the regular bootstrap method in IQ-TREE (1,000 replicates). Bootstrap values were only displayed when greater than 50%. The phylogenetic trees were visualized using Fig-Tree version 1.4.0. The phylogenetic tree match between RdRP and CP was estimated by the tanglegram algorithm in Dendroscope [43]. The genotype groupings for CaAV-1 and CTV were assessed using the Neighbor-Net approach in SplitsTree5 [44].

## Recombination analysis

Recombination was detected using both the RDP4 package [45] and by observing phylogenetic tree structural incongruities from different regions of the viral genome. The phylogenetic tree generated from the complete nucleotide and the amino acid sequences of the polyprotein (ORF1a and 1b), HSP70h and CP genes were used to compare the tree structural incongruities. Subsequently, the aligned nucleotide sequences were imported into RDP4 and analyzed using the RDP, GENECONV, Chimaera, MaxChi, BootScan, SiSan and 3Seq methods. The potential recombination events were considered significant only when supported by at least four methods with p-value $< 10^{-6}$. The recombination frequency of CTV and CaAV-1 was calculated based on the proportion of recombinants at a specific site among the total isolates, and the corresponding sequence identities were accessed using SimPlot analysis [46]. SNPs in coding regions were detected using the CLC Genomics Workbench 9.5.

## Supporting information

**S1 Fig. Phylogenetic analysis of the family *Closteroviridae* (left) and sequence similarity analysis for newly-identified closterovirids (right).** (A) ML tree derived from the HSP70 homolog gene of CiVB, CaAV-1, and CaAV-2, and representative members of the family *Closteroviridae*. Two plant HSP70 genes were used as an outgroup. Branch support was inferred by bootstrapping with 1,000 replicates. The scale bar represents the number of substitutions per site. Nucleotide and amino acid sequence identities of the different CiVB proteins for the most closely related closterovirids (B) and amino acid sequence identity of CaAV-1 and CaAV-2

proteins for the most closely related ampeloviruses (C).
(TIF)

**S2 Fig. Biological symptom analysis of citrus plants with CaAV-1 single infection.** (A) Leaf-roll symptoms on wild citrus sample YNL8-2. (B) Leaf margin becoming irregular and leaf blade upward or down curling symptoms on Morocco sour orange. (C) Boat-shaped leaf curling symptom on Duncan grapefruit. The right parts of panels B and C represent healthy controls.
(TIF)

**S3 Fig. Gene duplication analysis of closterovirids.** (A) Genomic locations and sequence alignment of p9b and p11, p34a and p34b of CiVB. The replicate genes are labeled in the same stripe shape in the CiVB genome, and the conserved sequences in alignment are labeled in red color. (B) ML tree derived from the CP, CPm, and p60~like genes of CiVB, CaAV-1, CaAV-2, and representative members of the family *Closteroviridae*. Two alphaviruses were used as an outgroup. Branch support was inferred by bootstrapping with 1,000 replicates. The scale bar represents the number of substitutions per site.
(TIF)

**S4 Fig. Multiple sequence alignment of closterovirids horizontally acquired genes.** Sequence alignment for closterovirids TLP (A), divalent metal cations transporter (B), and DEDDy 3'-5' exonuclease (C) genes with their respective homologs. CaAV-1, citrus associated ampelovirus 1; PAVA, pistachio ampelovirus A; FVA, fig virus A; FVB, fig virus B; OYLaV, Olive leaf yellowing-associated virus; PeVB, Persimmon virus B; AcV-1, Actinidia virus 1.
(TIF)

**S5 Fig. Phylogenetic analyses of CTV isolates.** ML trees reconstructed from the nucleotide sequences of the polyprotein (A), HSP70h (B), and CP (C). The values on each node are the percentages of 1,000 bootstrap replicates supporting the branch pattern. Scale bars represent numbers of substitutions per site. CTV isolates from wild citrus are in red.
(TIF)

**S6 Fig. Phylogenetic analyses of CaAV-1 isolates.** ML trees for CaAV-1 reconstructed from the complete genome sequence (A) and nucleotide sequence of the polyprotein (B), HSP70h (C), and CP (D). The values on each node are the percentages of 1,000 bootstrap replicates supporting the branch pattern. Scale bars represent numbers of substitutions per site. Different colors for clade names represent the origin of the samples: yellow, location 1; pink, location 2; blue, location 3.
(TIF)

**S7 Fig. Recombination and sequence variation analysis of CaAV-1 and CTV.** The left part represents two special examples of 3'-half genome recombination of CaAV-1. Transcriptome mapping of CaAV-1 isolates YNL8-21 (A), YNL8-22 (B), YNL8-71 (C), and YNL8-72 (D) identified in samples YNL8-2 (A and B) and YNL8-7 (C and D). The isolates YNL8-22 and YNL8-72 belong to genotype *A*; YNL8-21 and YNL8-71 belong to genotype *B*. Shared redundant reads are in yellow. The right part represents sequence variation schemes of CTV (E) and CaAV-1 (F). The levels of sequence variation per 1,000 nt in coding regions of CTV and CaAV-1 isolates are shown in heatmaps; means are shown in the line charts (top). RdRP, RNA-dependent RNA polymerase; HSP70h, heat shock protein 70 homolog; CP, major coat protein; CPm, minor coat protein.
(TIF)

**S1 Table. Sampling and sequencing information in this study.** The information concerning library construction and corresponding geographic locations, collection dates, and species of samples for each sequencing library.
(XLSX)

**S2 Table. *De novo* assembly information.** The corresponding reads numbers and average coverage of *de novo* assembly contigs in each sequencing library.
(XLSX)

**S3 Table. The presence of citrus closterovirids in wild citrus.** The positive samples and ratios of citrus closterovirids in wild citrus samples tested with RT-PCR.
(XLSX)

**S4 Table. Primers used in this study.** List of primers designed based on *de novo* assembly contigs for amplifying the full-genome sequence of CiVB, two genotypes of CaAV-1, and CaAV-2.
(XLSX)

**S1 Data. The relevant virus sequences data that acquired from wild citrus.**
(FA)

## Acknowledgments

We thank the late Dr. Ricardo Flores Pedauyé for his enthusiastic encouragement and inspirational guidance for our research. We thank Pedro Moreno for his critical reading and constructive comment on our manuscript. We thank LetPub (www.letpub.com) for its linguistic assistance during the preparation of this manuscript.

## Author Contributions

**Conceptualization:** Song Zhang, Guan-Zhu Han, Changyong Zhou, Mengji Cao.

**Data curation:** Qiyan Liu, Mengji Cao.

**Formal analysis:** Qiyan Liu, Song Zhang, Shiqiang Mei, Jianhua Wang, Mengji Cao.

**Funding acquisition:** Changyong Zhou, Mengji Cao.

**Investigation:** Qiyan Liu, Yan Zhou, Lei Chen.

**Project administration:** Changyong Zhou, Mengji Cao.

**Software:** Qiyan Liu, Jianhua Wang, Mengji Cao.

**Supervision:** Changyong Zhou, Mengji Cao.

**Writing – original draft:** Qiyan Liu.

**Writing – review & editing:** Qiyan Liu, Song Zhang, Guan-Zhu Han, Changyong Zhou, Mengji Cao.

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
