## [Decision Letter · Decision Letter 0]

22 Feb 2021

Dear Dr. Cao,

Thank you very much for submitting your manuscript "Viromics unveils extraordinary genetic diversity of the family Closteroviridae in wild citrus" for consideration at PLOS Pathogens. As with all papers reviewed by the journal, your manuscript was reviewed by members of the editorial board and by several independent reviewers. In light of the reviews (below this email), we would like to invite the resubmission of a significantly-revised version that takes into account the reviewers' comments.

Your paper has been assessed by two reviewers, both felt this was an important and interesting study, but reviewer 1 pointed out a number of major flaws in the paper that must be addressed before it can be acceptable for publication. Given the importance of the issue, I am advising you to revise the paper taking into account all of the comments. These need to be addressed in the Manuscript. I would also encourage you to find a colleague with expertise in virus evolution who could join you in this work as a co-author. Given the amount of revision required, should you choose to continue the paper will be reviewed again.

We cannot make any decision about publication until we have seen the revised manuscript and your response to the reviewers' comments. Your revised manuscript is also likely to be sent to reviewers for further evaluation.

Sincerely,

Marilyn J. Roossinck

Guest Editor

PLOS Pathogens

Peter Nagy

Section Editor

PLOS Pathogens

Kasturi Haldar

Editor-in-Chief

PLOS Pathogens

orcid.org/0000-0001-5065-158X

Michael Malim

Editor-in-Chief

PLOS Pathogens

orcid.org/0000-0002-7699-2064

Your paper has been assessed by two reviewers, both felt this was an important and interesting study, but reviewer 1 pointed out a number of major flaws in the paper that must be addressed before it can be acceptable for publication. Given the importance of the issue, I am advising you to revise the paper taking into account all of the comments. These need to be addressed in the Manuscript. I would also encourage you to find a colleague with expertise in virus evolution who could join you in this work as a co-author. Given the amount of revision required, should you choose to continue the paper will be reviewed again.

Reviewer's Responses to Questions

**Part I - Summary**

Reviewer #1: This work describes discovery of the CTV and three novel closteroviruses in the wild citrus growing in its natural mountainous habitat in China. Given the significant negative impact of CTV on citrus industry worldwide and the potential of new emerging virus infections to affect citrus growers, this work provides important insight into understanding ecology and evolution of citrus-infecting closteroviruses. Moreover, the genomic layout of the new closteroviruses described in this study significantly broadens understanding of genomic and evolutionary plasticity of this rapidly growing virus family and is likely to contribute to revision of closterovirus phylogeny and taxonomy.

In its present form, however, this manuscript needs to be significantly revised and re-written to address a number of the following major comments concerned with the quality of presented analyses and writing.

Reviewer #2: This is an innovative paper is providing molecular evidence for previously unrealized cases of infections of wild citrus relatives by two novel virus species belonging to ampelovirus genera and a third virus representing a possible new genus in the Closteroviridae family. Besides extending the citrus virus and the closteroviridae catalogs, this ms shows the significance of natural reserves as potential sources of disease agents endangering cultivated crop plants. Furthermore locating two and possibly three of the Closteroviridae family genera representatives within a limited area of undisturbed natural reserve could also indicate on the close association of the family origin with wild citrus On the other hand the finding of 5 out of the seven main CTV strains reported throughout the citrus world, associated with these wild citrus relatives either indicates , that CTV also originated in this area or that the native vegetation became infested by CTV strains carried by long distance flying vectors derived from long distance cultivated citrus hosting these isolates. For completing this paper the authors are suggested to include a supplemental map indicating on the distances of commercially cultivated citrus in the surrounding areas of the reserve.

**Part II – Major Issues: Key Experiments Required for Acceptance**

Reviewer #1: 1. The study would significantly benefit from providing data on: i) virus incidence (% of the wild plants infected with each of the 4 viruses); ii) occurrence of the mixed infections; iii) presence of disease symptoms in the screened wild plants and, most importantly, in the cultivated Morocco orange plants used for grafting. Although not strictly required, addition of such data would significantly expand the biological/plant pathological dimension of this work, which, at this point, is entirely focused on virus phylogenomics.

2. By and large, the data presentation and interpretation shows insufficient expertise of the authors in molecular biology of the closteroviruses and virus evolution in general. But a few telling examples are as follows: i) In Fig. 3, line 162 and elsewhere, the closterovirus ~60 KDa proteins are named ‘HSP90h’ (for homologs of HSP90 cellular molecular chaperones), a long obsolete notion refuted in the work by Napuli et al. (2003; PMID:12551975); ii) terms ‘intraspecies’ or ‘external genes’ are not in use in the field; iii) claiming that CiVB with monopartite genome is likely an ancestor of the criniviruses with split genomes is a run-away speculation that has no support in the data (in RdRP tree – Fig. 4 – CiVB is as divergent from criniviruses as it goes and so is CiVB CP in the same figure); iv) statement that some of the closterovirus genes that are not universally conserved in the family are under negative selection (e.g., lines 319-321) has no basis in presented data: gene gain and loss is one of the dominant processes in genome evolution that cannot possibly be branded as positive or negative selection; etc., etc.

3. The reliability of the conserved domain identification in this study is a suspect. In the lines 171-172, it is casually mentioned that CaAV-1 genome encodes two subunits of the DNA polymerase III: none of the RNA viruses known so far does encode any part of any DNA polymerase (there is no DNA phase in their replication cycles). If this is correct, such finding must be supported by identifying such domains in more than one isolate of this virus as well as presenting reliable, statistically robust multiple alignment of these subunits from CaAV-1 and diverse cellular organisms. Same is required for ‘divalent metal cations transport protein’ and ‘DEDDy 3’-5’ exonuclease (lines 176-178). If correct, the latter finding would be the only exonuclease encoded by RNA viruses in addition to those in coronaviruses, potentially a very significantly discovery suggestive of the proof-reading capability of this virus. Further, the lack of L-Pro in all three newly discovered viruses would be exceedingly surprising (if true) and more likely hints at rather low sensitivity of the employed search software.

4. The quality of language and writing in general are also significant problems. But a few beyond the pale examples are sentences in lines 18-23, 291, 336-338. The entire Discussion is excessively long and largely incoherent; the section ‘Unique mode…’ starting on line 236 belongs to Discussion in much briefer form (experimental confirmation of the lack of frameshift should be mentioned), many acronyms in the text were never spelled out etc., etc.

Reviewer #2: (No Response)

**Part III – Minor Issues: Editorial and Data Presentation Modifications**

Reviewer #1: (No Response)

Reviewer #2: See above

PLOS authors have the option to publish the peer review history of their article (what does this mean?). If published, this will include your full peer review and any attached files.

Reviewer #1: No

Reviewer #2: **Yes: **Bar-Joseph, Moshe
---

## [Decision Letter · Decision Letter 1]

24 Jun 2021

Dear Dr. Cao,

We are pleased to inform you that your manuscript 'Viromics unveils extraordinary genetic diversity of the family Closteroviridae in wild citrus' has been provisionally accepted for publication in PLOS Pathogens.

Best regards,

Marilyn J. Roossinck

Guest Editor

PLOS Pathogens

Peter Nagy

Section Editor

PLOS Pathogens

Kasturi Haldar

Editor-in-Chief

PLOS Pathogens

orcid.org/0000-0001-5065-158X

Michael Malim

Editor-in-Chief

PLOS Pathogens

orcid.org/0000-0002-7699-2064

Reviewer Comments (if any, and for reference):

Reviewer's Responses to Questions

**Part I - Summary**

Reviewer #1: This work provides major contributions to understanding origins and evolution of viruses affecting crop production in general and citrus in particular. Focused on closteroviruses, it also discovers novel members of this family, potentially at the genus level and novel virus-associated genes likely acquired via HGT from host of co-infecting or commensal microbes. One of these genes encoding thaumatin-like protein hints at recruitment of host defense proteins for virus functions common among diverse viruses, whereas another, a homolog of riboexonuclease, suggest a tantalizing possibility for the proof-reading role in the RNA replication process so far known only for coronaviruses. The data are solid, discussions comprehensive and the writing style is concise and clear. By and large, this reviewer is very impressed with authors bringing this thoroughly revised manuscript to qualitatively different level by improving both the style and the essence of the scientific narrative.

**Part II – Major Issues: Key Experiments Required for Acceptance**

Reviewer #1: None

**Part III – Minor Issues: Editorial and Data Presentation Modifications**

Reviewer #1: The only minor comment is the tendency of the authors to nearly exclusively cite review articles rather than original research that established critical aspects of closterovirus molecular biology and evolution.

PLOS authors have the option to publish the peer review history of their article (what does this mean?). If published, this will include your full peer review and any attached files.

Reviewer #1: **Yes: **Valerian V. Dolja

---

## [Editor Report · Acceptance letter]

7 Jul 2021

Dear Dr. Cao,

We are delighted to inform you that your manuscript, "Viromics unveils extraordinary genetic diversity of the family *Closteroviridae* in wild citrus," has been formally accepted for publication in PLOS Pathogens.

Best regards,

Kasturi Haldar

Editor-in-Chief

PLOS Pathogens

orcid.org/0000-0001-5065-158X

Michael Malim

Editor-in-Chief

PLOS Pathogens

orcid.org/0000-0002-7699-2064